# Targeting the Unfolded Protein Response with Natural Products: Therapeutic Potential in ER Stress-Related Diseases

**DOI:** 10.3390/ijms26188814

**Published:** 2025-09-10

**Authors:** Simona Martinotti, Elia Ranzato

**Affiliations:** DiSIT-Dipartimento di Scienze e Innovazione Tecnologica, University of Piemonte Orientale, Viale Teresa Michel 11, 15121 Alessandria, Italy; simona.martinotti@uniupo.it

**Keywords:** ER stress, unfolded protein response (UPR), natural products

## Abstract

This review delves into the intricate relationship between ER stress, the UPR, and human disease, with a specific focus on the therapeutic potential of natural products. We classify and discuss a wide range of natural compounds based on their unique mechanisms of action, whether they act as UPR inhibitors, activators, or indirectly alleviate ER stress by reducing oxidative burden or improving protein folding. By synthesizing the current literature, this review aims to provide a valuable resource for researchers and clinicians, highlighting the most promising natural products and their potential for development into novel therapeutic agents for treating pathologies driven by ER stress.

## 1. Introduction

The endoplasmic reticulum (ER), a vast and intricate network of membranes, serves as a crucial intracellular organelle responsible for a wide array of fundamental cellular processes. It is the main site for the synthesis of lipids, including phospholipids and steroids, and plays a central role in the biosynthesis of cholesterol [1,2].

Critically, the ER’s primary function is to serve as the cellular protein factory, where newly synthesized proteins are folded, modified, and assembled into their correct three-dimensional structures [3]. This delicate process is managed by a suite of ER-resident chaperones, such as BiP (also known as GRP78), and enzymes that ensure a rigorous quality control system. Additionally, the ER is the cell’s largest intracellular calcium reservoir, maintaining calcium homeostasis that is vital for regulating countless cellular signaling events, from gene expression to muscle contraction [4].

### 1.1. The Onset of Endoplasmic Reticulum (ER) Stress

The cell’s ability to function properly hinges on the ER’s capacity to handle the protein and lipid load it is tasked with. However, this capacity can be easily overwhelmed by various physiological and pathological conditions, including oxidative stress, viral infections, nutrient deprivation, hypoxia, or genetic mutations that lead to defective protein folding [5]. When the accumulation of unfolded or misfolded proteins in the ER lumen surpasses the organelle’s ability to process them, a state of ER stress is induced [6]. This condition represents a serious threat to cellular viability, as the buildup of misfolded proteins can lead to the formation of toxic aggregates and functional impairment of the ER itself [6].

To cope with ER stress, cells have evolved a sophisticated and highly conserved adaptive signaling pathway known as the unfolded protein response (UPR) [7]. The UPR is initiated by three key ER-resident transmembrane proteins that act as stress sensors: PERK, IRE1α, and ATF6 [1].

Initially, the UPR functions to restore homeostasis by attenuating global protein synthesis (via PERK), upregulating the expression of chaperones and protein degradation machinery (via IRE1 and ATF6), and enhancing the ER’s folding capacity. This protective phase is a crucial survival mechanism.

However, if ER stress is persistent or too severe for these adaptive measures to succeed, the UPR pivots to a pro-apoptotic program, primarily through the activation of the transcription factor CHOP, ultimately leading to programmed cell death to eliminate the damaged cell and prevent systemic harm.

The dysregulation of the UPR and chronic ER stress are now firmly established as pivotal drivers in the pathogenesis of numerous human diseases [5].

In neurodegenerative diseases such as Alzheimer’s and Parkinson’s, ER stress contributes to the aggregation of misfolded proteins (e.g., amyloid-beta and alpha-synuclein), which is a hallmark of these conditions [8,9].

In metabolic disorders like type 2 diabetes and obesity, ER stress in key metabolic organs such as the liver and pancreas impairs insulin signaling, leading to insulin resistance and beta-cell dysfunction [10].

Moreover, cancer cells often exploit the UPR to survive the challenging conditions of the tumor microenvironment, including hypoxia and nutrient deprivation, with ER stress promoting tumor growth, angiogenesis, and metastasis. Its involvement also extends to inflammatory diseases, cardiovascular pathologies, and viral infections, underscoring the universal importance of ER function in maintaining cellular health [11,12].

### 1.2. Rationale for Investigating Natural Products as Therapeutic Modulators

Given the central role of ER stress in disease, the UPR has emerged as a compelling therapeutic target [13]. This has spurred a growing interest in natural products—compounds derived from a rich diversity of sources including plants, fungi, and marine organisms. For millennia, these compounds have formed the basis of traditional medicine, and modern science is now validating their powerful and often pleiotropic biological activities [12,14]. Unlike many synthetic drugs that are designed to hit a single molecular target, natural products often possess a range of bioactivities—such as antioxidant, anti-inflammatory, and anti-cancer effects—that can modulate complex disease pathways like the UPR in a more holistic manner. Their vast chemical diversity provides an unparalleled reservoir of novel chemical scaffolds for drug discovery, offering the potential to develop new, effective, and potentially safer agents for modulating ER stress.

Despite the broad recognition of the UPR’s therapeutic potential and the rich diversity of natural products, the specific mechanisms by which many of these compounds modulate the UPR remain underexplored. While some studies have focused on single compounds or isolated pathways, there is a notable gap in the literature regarding a systematic, multi-faceted analysis of natural product derivatives and their integrated effects on the full spectrum of UPR signaling branches (PERK, IRE1α, and ATF6).

This study addresses this critical gap by providing a comprehensive investigation into the molecular mechanisms through which natural compounds modulate the UPR.

## 2. The Endoplasmic Reticulum and the Unfolded Protein Response (UPR)

The endoplasmic reticulum (ER) is a highly dynamic and responsive organelle, whose functional integrity is paramount for cell survival.

Its vast network of membranes maintains a distinct internal environment, known as the ER lumen, which is a finely tuned system for protein folding and modification. The stability of this environment, or ER homeostasis, is continuously monitored. However, this delicate balance can be readily disrupted by a variety of internal and external factors, leading to a state of ER stress [15].

### 2.1. ER Homeostasis and Stress Inducers

ER stress can be triggered by any condition that leads to an accumulation of unfolded or misfolded proteins in the ER lumen. These inducers are diverse and impact different aspects of ER function. Hypoxia and nutrient deprivation are common stressors, as they deplete the cell’s ATP reserves, which are essential to power the chaperones and enzymes that assist in protein folding [16]. Oxidative stress, caused by an excess of reactive oxygen species (ROS), can directly damage proteins and the ER membrane, impairing function [11].

The ER is also the cell’s largest intracellular calcium store, and a disruption in its calcium balance, often due to the malfunction of SERCA pumps or ion channels, can severely impair the activity of calcium-dependent chaperones [17]. Additionally, certain genetic mutations lead to the production of proteins that are intrinsically prone to misfolding, such as the mutant protein in cystic fibrosis [18]. Finally, viral infections can overwhelm the ER by forcing it to synthesize a massive amount of viral proteins, leading to a folding bottleneck [19].

### 2.2. Key UPR Signaling Pathways

To counteract ER stress and restore homeostasis, cells activate the unfolded protein response (UPR), a sophisticated signaling network regulated by three key ER-resident transmembrane sensors: PERK, IRE1α, and ATF6 (see also Figure 1).

#### 2.2.1. The PERK Pathway

The PERK (PKR-like ER Kinase) pathway is a rapid-acting response to ER stress. Under normal conditions, the ER chaperone BiP (also known as GRP78) binds to and keeps PERK in an inactive state [5]. When unfolded proteins accumulate, BiP preferentially binds to them, releasing PERK. This release allows PERK to dimerize and autophosphorylate, activating its kinase domain. Activated PERK then phosphorylates the alpha subunit of eukaryotic initiation factor 2 (eIF2α). Phosphorylation of eIF2α leads to a global attenuation of protein synthesis, a critical survival mechanism that reduces the protein folding load on the stressed ER. While general translation is suppressed, this phosphorylation paradoxically allows for the selective translation of specific mRNAs, most notably that of the transcription factor, activating transcription factor 4 (ATF4). ATF4 then enters the nucleus to upregulate the expression of genes involved in amino acid metabolism and the antioxidant response, as well as pro-apoptotic genes like CHOP [5].

#### 2.2.2. The IRE1α Pathway

The IRE1α (inositol-requiring enzyme 1 alpha) pathway is another central branch of the UPR [20]. Like PERK, IRE1α is kept inactive by BiP. Upon stress, its release from BiP triggers its oligomerization and autophosphorylation, activating its dual function as both a kinase and a site-specific endoribonuclease. The key event in this pathway is the splicing of a specific 26-nucleotide intron from the X-box binding protein 1 (XBP1) mRNA. This splicing event shifts the mRNA reading frame, producing a more stable and powerful transcription factor, XBP1s.

XBP1s translocates to the nucleus where it drives the transcription of a wide range of genes dedicated to expanding the ER’s folding capacity, including those for ER chaperones, components of the ER-associated degradation (ERAD) machinery, and enzymes involved in lipid biosynthesis to expand the ER membrane. IRE1α also has a second nuclease function, known as IRE1αdependent mRNA decay (RIDD), which degrades specific mRNAs to further reduce the protein load and dampen inflammatory responses [21].

#### 2.2.3. The ATF6 Pathway

The ATF6 (activating transcription factor 6) pathway represents the third branch of the UPR [20]. ATF6 is a transmembrane protein that, upon ER stress, is released from BiP and moves from the ER to the Golgi apparatus. In the Golgi, it undergoes a process of regulated intramembrane proteolysis, where it is sequentially cleaved by two proteases (S1P and S2P). This cleavage releases the N-terminal fragment of ATF6, referred to as ATF6 (N), which is the active transcription factor. ATF6 (N) then migrates to the nucleus to induce the transcription of genes encoding ER chaperones (e.g., GRP78, GRP94) and ERAD components, acting in synergy with the IRE1α pathway to bolster the ER’s capacity.

### 2.3. The Dual Nature of the UPR

The UPR is characterized by its remarkable dual role as a survival and a death pathway [22]. In conditions of mild or transient ER stress, the UPR functions primarily as a survival mechanism. Its initial adaptive responses—the translational attenuation by PERK, the chaperone upregulation by IRE1α and ATF6, and the enhancement of ERAD—are all designed to restore homeostasis and protect the cell from damage.

However, if the stress is chronic or severe, these protective measures prove insufficient. In this scenario, the UPR pivots to a pro-apoptotic program. This transition is mediated by the sustained activation of certain UPR factors, most notably the pro-apoptotic transcription factor CHOP, whose expression is promoted by both the PERK and ATF6 pathways [23].

CHOP then suppresses the expression of anti-apoptotic proteins and induces pro-apoptotic ones, initiating the cell death cascade. This shift from an adaptive to an apoptotic response is a critical mechanism for eliminating irreversibly damaged cells, thereby preventing systemic harm to the organism [24].

## 3. ER Stress in Disease Pathogenesis

The intricate link between endoplasmic reticulum (ER) stress and human disease is a cornerstone of modern pathology. While the UPR is an essential protective mechanism, its prolonged or dysregulated activation due to chronic ER stress is a key driver of cellular dysfunction, maladaptation, and cell death. The failure to resolve ER stress contributes to the initiation and progression of a wide range of human pathologies, from neurodegeneration to cancer [25]. The following sections detail the involvement of ER stress in major disease categories, highlighting the specific mechanisms through which it contributes to pathogenesis (see also Table 1).

### 3.1. Cancer

Cancer cells thrive in harsh tumor microenvironments characterized by hypoxia, nutrient deprivation, and metabolic stress, all of which are potent inducers of ER stress. Unlike healthy cells, cancer cells often hijack and manipulate the UPR to their advantage. A constitutively active UPR allows these cells to adapt to stress, promoting cell survival, resistance to apoptosis, and chemoresistance [26]. The PERK-eIF2α-ATF4 pathway, for instance, is frequently activated in tumors to attenuate protein synthesis and enable survival under nutrient-deprived conditions [27]. At the same time, ATF4 promotes the transcription of genes involved in metabolism and antioxidant defense [28]. Furthermore, the IRE1α pathway plays a crucial role in enabling cancer cells to adapt their metabolism and can activate pro-survival pathways like NF-κB, promoting angiogenesis and facilitating metastasis and tumor spread [29]. This manipulation of ER stress pathways highlights their potential as novel therapeutic targets in oncology.

### 3.2. Neurodegenerative Diseases

A hallmark of many neurodegenerative diseases, such as Alzheimer’s disease, Parkinson’s disease, and Huntington’s disease, is the accumulation of misfolded and aggregated proteins, a process that is inextricably linked to ER stress [30].

In Alzheimer’s, the accumulation of amyloid-beta (Aβ) peptides and neurofibrillary tangles (composed of hyperphosphorylated tau protein) directly overloads the ER’s protein-folding machinery. This chronic ER stress, in turn, exacerbates the misfolding and aggregation of these proteins, creating a vicious cycle that leads to synaptic dysfunction and progressive neuronal cell death [8].

In Parkinson’s, the aggregation of α-synuclein is a major stressor. This accumulation disrupts ER function and impairs the ubiquitin–proteasome system, a key part of the ER-associated degradation (ERAD) pathway, leading to a buildup of toxic proteins and increased ER stress [31].

In Huntington’s, the mutant Huntingtin protein undergoes misfolding and aggregation, triggering ER stress that culminates in neuronal apoptosis in specific brain regions. The chronic activation of the pro-apoptotic arm of the UPR, particularly via CHOP, is a major contributor to the progressive loss of neuronal function characteristic of these diseases [32].

### 3.3. Metabolic Disorders

ER stress is a key pathological feature of major metabolic diseases, acting as a bridge between metabolic imbalances and cellular dysfunction. This state of cellular imbalance is now considered a central hub linking over-nutrition and obesity to the development of insulin resistance and type 2 diabetes (T2D) [33].

In type 2 diabetes, chronic ER stress in insulin-producing pancreatic β-cells impairs their function and ultimately leads to apoptosis via the CHOP pathway, contributing to the decline in insulin production. In parallel, ER stress in the liver and adipose tissue activates inflammatory kinases, such as JNK, which phosphorylate insulin receptor substrates, blocking insulin signaling and inducing a state of insulin resistance [33].

In obesity and fatty liver disease, the excess nutrient load and lipid accumulation, particularly of saturated fatty acids, directly induce ER stress [34]. Saturated fatty acids, unlike their unsaturated counterparts, are potent inducers of ER stress because they can integrate into the ER membrane, altering its fluidity and disrupting the activity of membrane-associated folding enzymes and chaperones [35]. This stress drives inflammatory responses and promotes the accumulation of lipids in hepatocytes, leading to hepatic steatosis (fatty liver). The activated UPR promotes the release of pro-inflammatory cytokines, further exacerbating liver damage and contributing to the transition from simple steatosis to more severe disease and ultimately, fibrosis and cirrhosis. Targeting ER stress pathways is therefore a promising therapeutic strategy to break the link between metabolic dysfunction and organ-specific pathology [36].

### 3.4. Inflammatory and Autoimmune Diseases

The relationship between ER stress and inflammation is bidirectional and central to the pathogenesis of many inflammatory and autoimmune conditions. This bidirectional link forms a powerful positive feedback loop: ER stress triggers inflammation, and inflammatory cytokines, in turn, can induce ER stress. This vicious cycle is a key driver of chronic pathology [37]. ER stress signaling can activate the inflammatory transcription factor NF-κB, leading to the production of pro-inflammatory cytokines such as IL-6 and TNF-α. This link is critically implicated in conditions like inflammatory bowel disease (IBD) and rheumatoid arthritis, where persistent, low-grade inflammation is a key driver of tissue damage and disease progression. For instance, in IBD, an impaired intestinal epithelial barrier can lead to a state of chronic ER stress, which then fuels the inflammatory response by activating NF-κB in epithelial and immune cells, ultimately compromising the gut’s integrity [38]. In immune cells, such as B cells, the UPR is naturally activated to support the massive production of antibodies [39]. However, its dysregulation can lead to the production of autoantibodies, contributing to autoimmune pathology. The IRE1α pathway, in particular, plays a crucial role in coordinating both adaptive immunity and the inflammatory response [40].

### 3.5. Cardiovascular Diseases

ER stress significantly impacts the cardiovascular system, contributing to a range of pathologies [41].

In atherosclerosis, ER stress in macrophages and endothelial cells promotes lipid uptake, triggers inflammatory signaling, and contributes to the formation of lipid-rich foam cells, a key step in atherosclerotic plaque development [42].

During myocardial ischemia/reperfusion injury, the cycles of hypoxia and reoxygenation induce severe ER stress, leading to cardiomyocyte dysfunction and apoptosis, a major cause of heart failure [43].

ER stress can also disrupt calcium handling within the cell, which is essential for cardiac contraction and rhythm, potentially contributing to arrhythmias and a decline in overall cardiac function. The UPR’s adaptive mechanisms can initially protect the heart, but prolonged stress ultimately leads to pathological remodeling and heart failure [44].

### 3.6. Respiratory Diseases

The ER is a vital organelle in respiratory cells, particularly in airway epithelial cells, alveolar cells, and immune cells, where it is responsible for the synthesis and folding of a vast array of proteins essential for lung function, including mucins, surfactants, and secreted immunoglobulins.

When this delicate process is disrupted, a state of ER stress ensues, triggering the UPR to restore homeostasis. The UPR and its pathological dysregulation are now recognized as key contributors to the pathogenesis of several major respiratory diseases.

In chronic obstructive pulmonary disease (COPD), ER stress is a central mechanism linking cigarette smoke exposure to airway inflammation and emphysema. Inhaled cigarette smoke contains numerous toxic compounds that directly damage ER membranes and disrupt calcium homeostasis, leading to the accumulation of misfolded proteins. This chronic ER stress triggers a sustained and maladaptive UPR. The persistent activation of CHOP, a pro-apoptotic transcription factor, drives the programmed cell death of airway epithelial and alveolar cells, contributing to the progressive destruction of lung parenchyma characteristic of emphysema. Furthermore, the smoke-induced UPR activates inflammatory signaling pathways, such as NF-κB, which amplifies airway inflammation and mucus hypersecretion, which are key features of chronic bronchitis [45,46].

Understanding ER stress and UPR signaling is crucial for developing targeted therapies to restore cellular homeostasis and mitigate disease progression.

### 3.7. Viral Infections

Viruses are masters at manipulating host cell machinery for their own replication. Many viruses, including hepatitis C and SARS-CoV-2, rely extensively on the host’s ER to synthesize and fold their proteins [47]. This massive protein load can directly induce ER stress. Viruses, in turn, exploit UPR signaling pathways to enhance their replication and evade the host’s antiviral immune response. By understanding how viruses manipulate ER stress, researchers can identify new therapeutic targets to interfere with viral life cycles.

The outcome of ER stress during a viral infection is a delicate balance: the UPR can activate antiviral responses and apoptosis, but it can also be co-opted by the virus to promote its survival and replication [48].

**Table 1 ijms-26-08814-t001:** This table provides a concise overview of the role of ER stress in the pathogenesis of various human diseases. It highlights how the dysregulation of the UPR, an adaptive mechanism, can become a central driver of cellular dysfunction and disease progression in different pathological contexts.

Disease	Role of Endoplasmic Reticulum Stress
**Cancer**	Cancer cells hijack the UPR to promote survival, apoptosis resistance, and chemoresistance. The PERK-eIF2α-ATF4 and IRE1α pathways are often activated to adapt to the hostile tumor environment [27].
**Neurodegenerative** **Diseases**	Chronic accumulation of misfolded proteins (e.g., Aβ, α-synuclein, mutant Huntingtin protein) overloads the ER, triggering stress. The activation of the pro-apoptotic CHOP arm contributes to progressive neuronal cell death [30].
**Metabolic Disorders (Type 2 Diabetes, Obesity)**	In pancreatic β-cells, ER stress leads to dysfunction and apoptosis. In the liver and adipose tissue, it induces inflammation and insulin resistance by activating kinases like JNK [33].
**Inflammatory and Autoimmune Diseases**	ER stress activates the NF-κB transcription factor, leading to the production of pro-inflammatory cytokines. UPR dysregulation in immune cells can contribute to the production of autoantibodies [37].
**Cardiovascular Diseases**	Promotes atherosclerotic plaque formation and contributes to cardiomyocyte dysfunction and apoptosis during ischemia/reperfusion. It can also impair calcium handling and cardiac function [41].
**Respiratory Diseases**	Cigarette smoke-induced ER stress activates the PERK-CHOP pathway [45].
**Viral Infections**	Viruses exploit the ER for their protein synthesis, inducing ER stress. They manipulate UPR signaling pathways to enhance their replication and evade the host’s immune response [47].

## 4. Natural Products Modulating ER Stress

The intricate link between ER stress and numerous pathologies has made the UPR an attractive therapeutic target. In this context, natural products, with their vast chemical diversity and pleiotropic bioactivities, have emerged as a promising avenue for therapeutic development [12]. These compounds offer a potential therapeutic strategy by either directly alleviating stress or by rebalancing the UPR towards its adaptive, pro-survival functions. This chapter details the classification of key natural compounds and explores their specific, often multi-faceted, mechanisms of action in modulating ER stress pathways.

### 4.1. Mechanisms of Action on ER Stress

Natural products influence ER stress through a variety of distinct mechanisms, often targeting multiple points within the UPR network to restore cellular homeostasis (see Table 2).

Direct Attenuation of ER Stress: Some natural compounds act as chemical chaperones, small molecules that can stabilize protein conformation and assist in protein folding within the ER lumen. This action directly prevents the accumulation of misfolded proteins and reduces the overall stress load. Others possess potent antioxidant properties, scavenging reactive oxygen species (ROS) that can damage ER proteins and membranes, thereby eliminating a major inducer of ER stress [49]. This is often achieved by activating the Nrf2 pathway, a master regulator of antioxidant genes. Additionally, some compounds can restore calcium homeostasis by modulating the activity of SERCA pumps, ensuring a stable environment for calcium-dependent folding enzymes.

Modulation of UPR Branches: This is a highly specific mechanism involving the direct influence on UPR’s signaling pathways. Certain natural products can inhibit harmful UPR pathways, for example, by blocking the kinase activity of PERK or the splicing activity of IRE1α [50]. Conversely, some compounds can selectively activate beneficial UPR pathways, such as promoting IRE1α/XBP1s or ATF6 activation to increase the transcription of protective genes that expand the ER’s folding capacity and enhance ER-associated degradation (ERAD).

Enhancing ER-Associated Degradation (ERAD): The ERAD pathway is the cell’s “waste disposal” system for terminally misfolded proteins. By upregulating components of this machinery, such as E3 ubiquitin ligases, some natural products help the cell efficiently clear these toxic proteins from the ER. This action directly reduces the stress signal and is a crucial step in restoring ER homeostasis.

### 4.2. Key Examples of Natural Products and Their Effects

#### 4.2.1. Flavonoids and Polyphenols

Curcumin: This well-studied polyphenol, derived from turmeric, is a potent modulator of ER stress with highly context-dependent effects [51].

In various cancer models, curcumin can inhibit the pro-survival PERK-eIF2α-ATF4 pathway, thereby promoting apoptosis and enhancing the efficacy of conventional therapies [52].

In contrast, in neurodegenerative and ischemic models, it exhibits neuroprotective effects by reducing oxidative stress and inflammation, indirectly alleviating the ER stress burden [53]. Its anti-inflammatory action is also linked to the inhibition of the NF-κB pathway, a downstream target of UPR signaling [54].

While much of the evidence for its ER stress-modulating effects comes from preclinical studies, its established anti-inflammatory and antioxidant properties have led to numerous clinical trials. These trials, often in the context of arthritis, metabolic syndrome, and inflammatory bowel diseases, show curcumin’s potential to reduce systemic inflammation and oxidative stress, both of which are key drivers of ER stress. This clinical evidence, while not always directly measuring UPR markers, supports the hypothesis that curcumin’s broader therapeutic benefits are, in part, mediated by its ability to restore cellular homeostasis [55].

Resveratrol: Found in grapes and berries, this stilbenoid is recognized for its beneficial effects on metabolic and cardiovascular diseases [56]. Resveratrol can alleviate ER stress by activating the SIRT1 pathway, a key deacetylase that enhances protein folding and reduces oxidative stress [57]. Its ability to mitigate ER stress in adipose tissue and the liver contributes to improved insulin sensitivity and reduced lipid accumulation [58]. Its activation of AMPK further promotes cellular energy homeostasis and autophagy, a process that helps to clear damaged organelles and misfolded proteins, thereby relieving ER stress [59]. While many of its direct ER stress-modulating effects have been observed in animal and in vitro models, resveratrol has been the subject of several clinical trials [60]. For example, studies have investigated its impact on insulin sensitivity, inflammation markers (such as C-reactive protein), and oxidative stress in postmenopausal women with high BMI and in elderly populations [61]. These trials, although not always focused on ER stress as a primary endpoint, provide important human data that supports the cellular mechanisms observed in preclinical studies.

Epigallocatechin-3-Gallate (EGCG): A major flavonoid in green tea, EGCG is known for its potent antioxidant and anti-inflammatory properties [62]. It can act as a chemical chaperone, preventing the aggregation of misfolded proteins by binding to them.

EGCG also modulates the UPR by inhibiting the PERK-eIF2α pathway and promoting the IRE1α/XBP1s branch, protecting cells from stress-induced apoptosis [63]. Its anti-inflammatory effects also play a crucial role in indirectly reducing ER stress.

Recent work [64] has shown that in malignant mesothelioma cells, EGCG re-calibrates a pro-survival UPR into a pro-apoptotic one, highlighting its potential to overcome chemoresistance.

The broad health benefits of green tea and its extracts, including cardiovascular and cognitive health, are well-documented in human studies [65]. Although direct clinical trials specifically on EGCG’s modulation of ER stress are limited, its proven anti-inflammatory, antioxidant, and metabolic effects provide a strong rationale for its therapeutic application in ER stress-related diseases.

Quercetin: This ubiquitous flavonoid primarily reduces ER stress through its potent antioxidant properties. By scavenging ROS, it directly eliminates a major stress inducer [66]. It can also suppress the pro-apoptotic CHOP pathway and modulate inflammatory signaling by inhibiting factors like NF-κB, contributing to its protective effects in various inflammatory and cardiovascular disease models.

Research indicates that quercetin specifically attenuates ER stress by modulating the PERK-eIF2α pathway, often by inhibiting the phosphorylation of PERK and its downstream target eIF2α [66]. This prevents the accumulation of ATF4 and CHOP, which are key drivers of apoptosis under chronic ER stress.

In human studies, quercetin has been investigated for its effects on inflammation, cardiovascular risk factors, and immune function [67]. The results from these trials, which consistently show reduced markers of oxidative stress and inflammation, corroborate the preclinical findings that its cellular benefits are directly tied to alleviating underlying stressors like ER stress.

Oleuropein: A key polyphenol found in olive leaves, oleuropein is a powerful antioxidant. It has been shown to reduce ER stress in neurodegenerative and cardiovascular models by scavenging free radicals and improving the function of cellular protein degradation systems [68]. Its ability to protect neurons from stress-induced cell death makes it a subject of significant interest [69]. The benefits of olive oil and Mediterranean diets, which are rich in oleuropein, on cardiovascular health and inflammation are widely supported by human population studies and clinical trials, providing strong indirect evidence for its cellular effects [70].

Baicalein (5,6,7-trihydroxyflavone) is a flavonoid, that exhibits significant biological activities, including antioxidant, anti-inflammatory, and neuroprotective properties [71]. Baicalein is predominantly sourced from the roots of plants belonging to the genus Scutellaria, commonly known as skullcap. Baicalein acts as a sophisticated modulator, capable of differentially influencing the UPR branches based on the specific cellular context, dose, and pathological state. This dual nature positions baicalein as a promising, yet challenging, candidate for UPR-targeted therapies.

Baicalein’s protective effect is often linked to its ability to inhibit PERK phosphorylation, thereby preventing the downstream cascade that leads to eIF2α phosphorylation and subsequent CHOP upregulation. By mitigating the pro-apoptotic arm of the PERK pathway, baicalein helps preserve cell viability and homeostasis [72]. This action is particularly relevant in conditions where ER stress is a key driver of pathology, such as in heart failure and neuroinflammation, where baicalein’s effects are often associated with reduced oxidative stress and improved cell survival [73].

#### 4.2.2. Alkaloids and Saponins

Berberine: An alkaloid used in traditional medicine, berberine has significant effects on metabolic ER stress. It primarily acts by activating the AMPK pathway, a master regulator of cellular energy homeostasis [74]. This activation leads to a cascade of events that reduce ER stress in the liver and adipose tissue, resulting in improved glucose and lipid metabolism, making it a promising candidate for treating type 2 diabetes and fatty liver disease [75]. The activation of AMPK by berberine also promotes autophagy, a process that helps clear damaged organelles and misfolded proteins, further relieving ER stress.

Berberine has been the subject of numerous clinical trials, particularly for its potent effects on blood sugar and lipid profiles in diabetic patients. Clinical studies have shown that berberine is as effective as metformin, a first-line drug for type 2 diabetes, in improving glycemic control, and its therapeutic action is thought to be strongly linked to its ability to attenuate the metabolic ER stress that underlies insulin resistance [76].

Piperine: This alkaloid from black pepper has been shown to exert anti-inflammatory and antioxidant effects. Studies indicate that piperine can modulate ER stress pathways, particularly in metabolic disorders, by reducing inflammation and oxidative stress, thereby protecting cells from damage and improving metabolic function [77]. Its anti-inflammatory action is also linked to the inhibition of NF-κB and a reduction in the production of pro-inflammatory cytokines [78]. While clinical trials specifically focused on piperine’s effect on ER stress are scarce, its role as a bioavailability enhancer for other compounds (like curcumin) is well-established in human studies [79]. Its anti-inflammatory and antioxidant effects have been documented in various in vivo and in vitro models, providing a strong basis for future clinical investigation into its direct effects on ER stress-related pathologies [80].

Ginsenosides: The primary active components of ginseng, these saponins are known for their neuroprotective and anti-diabetic effects. Ginsenosides have been shown to protect cells from ER stress-induced apoptosis by inhibiting the PERK-eIF2α-CHOP pathway, while also promoting beneficial cellular antioxidant responses. Their ability to modulate multiple stress pathways simultaneously makes them promising candidates for treating a variety of ER stress-related pathologies [81].

Astragaloside IV is a major active saponin derived from the root of Astragalus membranaceus, a plant widely used in traditional Chinese medicine [82]. Research has demonstrated that astragaloside IV can effectively alleviate ER stress by modulating the UPR pathways. Its primary mechanism appears to involve the suppression of the pro-apoptotic branches of the UPR. By inhibiting the phosphorylation of PERK and eIF2α, astragaloside IV reduces the expression of the pro-apoptotic factor CHOP, thereby preventing ER stress-induced cell death [83]. Furthermore, it has been shown to enhance the adaptive UPR by promoting the expression of ER chaperones, such as GRP78/BiP, which helps improve protein-folding capacity and maintain cellular homeostasis [84,85]. This dual action—inhibiting pro-death signaling while supporting pro-survival mechanisms—positions astragaloside IV as a promising therapeutic agent for diseases where chronic ER stress is a key pathological factor, including cardiovascular diseases, diabetes, and neurodegenerative disorders.

#### 4.2.3. Terpenoids and Other Compounds

Honokiol: A lignan isolated from the magnolia bark, honokiol possesses potent anti-cancer and anti-inflammatory properties [86]. Research suggests that honokiol can inhibit the UPR, specifically by targeting the IRE1α pathway, which in turn leads to the suppression of pro-survival signals in cancer cells, highlighting its potential as a targeted therapeutic agent [87].

Ursolic Acid: A triterpenoid found in the peels of many fruits (e.g., apples), ursolic acid has demonstrated significant anti-inflammatory and anti-cancer effects [88]. It has been shown to alleviate ER stress by downregulating the expression of pro-apoptotic UPR proteins, such as CHOP, and by promoting the expression of protective genes. Its ability to inhibit ER stress-induced apoptosis in various cell types highlights its potential as a broad-spectrum therapeutic agent [89,90].

Camphene is a monoterpene and a major component of essential oils derived from plants such as camphor, turpentine, and fir [91]. Preclinical studies have shown that camphene can induce mild to moderate ER stress in specific cell types, which in some cases, triggers an adaptive UPR. This activation is thought to be mediated by the induction of chaperone proteins and the selective modulation of UPR signaling branches. For example, some evidence suggests that camphene can activate the pro-survival IRE1α pathway, helping cells to adapt to and mitigate ER stress. In other contexts, particularly in cancer cells, camphene-induced ER stress can be leveraged to drive apoptosis by pushing the UPR towards its pro-death signaling, indicating a potential dual role similar to other UPR modulators [92]. These findings highlight camphene as a promising natural compound that can be investigated for its precise mechanisms of action within the ER-UPR axis, particularly in the context of its reported anti-inflammatory and anti-proliferative effects.

Sulforaphane: This isothiocyanate, found in cruciferous vegetables, acts as a potent inducer of the adaptive UPR [93]. It primarily functions by activating the Nrf2 pathway by binding to and inactivating its repressor protein, Keap1 [71]. This allows Nrf2 to translocate to the nucleus and drive the transcription of a whole suite of antioxidant and detoxifying genes, thereby indirectly alleviating the burden on the ER and promoting cell survival. Sulforaphane has been tested in several clinical trials [72]. For example, a Phase 1 clinical trial is investigating its safety and effects on markers of oxidative stress and inflammation in patients with chronic kidney disease. This type of research is crucial as it translates the profound cellular effects of sulforaphane, especially its Nrf2 activation, into measurable clinical benefits.

**Table 2 ijms-26-08814-t002:** This table summarizes the primary mechanisms of action of key natural compounds in modulating ER stress and inflammation. The compounds are categorized by their main functions, including inhibition of inflammatory pathways, antioxidant properties, and specific modulation of different branches of the UPR.

Natural Compound		Primary Mechanisms of Action
Curcumin	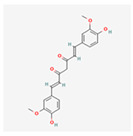	Inhibits the pro-survival PERK-eIF2α-ATF4 pathway, reduces oxidative stress and inflammation by blocking NF-κB [52].
Resveratrol	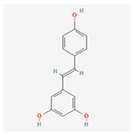	Activates the SIRT1 and AMPK pathways to improve protein folding and energy homeostasis, acts as an antioxidant [58].
EGCG	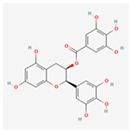	Acts as a chemical chaperone, inhibits the PERK-eIF2α pathway and promotes IRE1α/XBP1s, has strong antioxidant and anti-inflammatory properties [63].
Quercetin	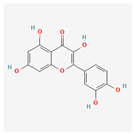	Potent antioxidant, suppresses the pro-apoptotic CHOP pathway, and modulates inflammatory signaling by inhibiting NF-κB [66].
Oleuropein	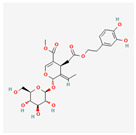	Powerful antioxidant that reduces ER stress and protects cells from free radicals [68].
Baicalein	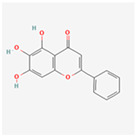	Alleviates ER stress and modulates UPR based on context (pro-survival vs. pro-apoptotic) [73].
Berberine	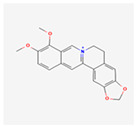	Activates the AMPK pathway, which reduces metabolic ER stress and improves glucose and lipid metabolism [74,76].
Ginsenosides	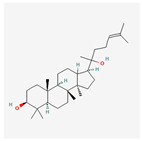	Protects cells from ER stress-induced apoptosis by inhibiting the PERK-eIF2α-CHOP pathway [81].
Piperine	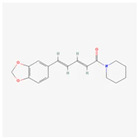	Exerts anti-inflammatory and antioxidant effects, modulates ER stress pathways, and inhibits NF-κB [80].
Astragaloside IV	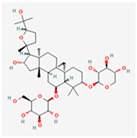	Inhibits pro-apoptotic pathways, enhances pro-survival GRP78/BiP expression [94].
Honokiol	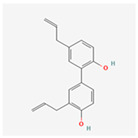	Inhibits the IRE1α branch of the UPR, suppressing pro-survival signals in cancer cells [87].
Ursolic Acid	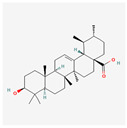	Alleviates ER stress by down-regulating pro-apoptotic UPR proteins and promoting the expression of protective genes [89].
Camphene	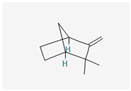	Induces ER stress, modulates UPR [92].
Sulforaphane	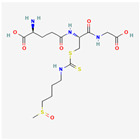	Activates the Nrf2 pathway, a master regulator of antioxidant and detoxifying genes, thereby indirectly alleviating ER stress [72].

## 5. Challenges and Future Directions

The investigation of natural products as modulators of ER stress presents both significant promise and notable challenges. While the potential for novel therapeutics is vast, a number of hurdles must be overcome to effectively translate preclinical findings into clinical applications (see also Table 3).

### 5.1. Challenges in Research and Development

The journey from a plant-derived compound to a clinically viable drug is complex, and several inherent factors complicate this process specifically for natural ER stress modulators. One of the most significant obstacles is a fundamental issue of bioavailability and pharmacokinetics. Many of these potent compounds, particularly polyphenols, were not evolved to be absorbed by the human digestive system in therapeutic doses [56]. They often suffer from poor oral absorption, are rapidly metabolized in the liver, and fail to reach target tissues in sufficient concentrations. This means that even a compound that shows remarkable efficacy in a Petri dish may have a negligible effect within the complex environment of a living organism, a major constraint on therapeutic development.

The use of natural extracts introduces significant variability, as their chemical composition and potency can change based on the plant’s origin, growing conditions, harvest time, and extraction methods [95]. This lack of consistency makes it difficult to conduct reproducible research, which is a fundamental requirement for the rigorous validation demanded in drug development.

Furthermore, the very strength of many natural compounds—their complexity of action—can also be a weakness. While their multi-target effects can offer a holistic therapeutic benefit, they also make it scientifically challenging to pinpoint the precise mechanism of action.

This complexity hinders the development of targeted, specific therapies, as it can be difficult to determine which of the many pathways influenced by a compound is most critical for its therapeutic effect.

Finally, a major bottleneck is the scarcity of robust human clinical trials [82]. Despite a wealth of compelling preclinical data, there is a significant gap in large-scale human studies investigating natural products for ER stress-related diseases. This is often a direct consequence of the aforementioned challenges, combined with a lack of dedicated funding and clear regulatory pathways for compounds that do not fit the traditional single-target synthetic drug model.

### 5.2. Future Perspectives

Addressing these challenges requires a concerted, multi-disciplinary effort. The future of natural product research as a therapeutic strategy for ER stress hinges on several key directions aimed at overcoming these barriers.

Advanced technologies in novel natural product discovery are moving the field beyond well-known compounds. High-throughput screening, coupled with computational biology and genomics, will allow researchers to more efficiently screen vast libraries of natural products from diverse sources to identify new lead molecules with more favorable properties.

The next logical step involves in-depth structure–activity relationship (SAR) studies [83]. By systematically modifying the chemical structure of a lead compound, researchers can identify the specific functional groups responsible for its therapeutic effect. This process allows for the development of semi-synthetic derivatives with enhanced potency, selectivity, and stability, paving the way for more effective and safer drugs.

One of the most promising solutions to the bioavailability challenge lies in advanced delivery systems, particularly in the field of nanotechnology [84]. Encapsulating natural products in nanoparticles, liposomes, or micelles can significantly improve their absorption, protect them from rapid degradation, and enable targeted delivery to specific tissues or cells, such as those in the brain or pancreas. This targeted approach could drastically increase their therapeutic efficacy while minimizing off-target side effects.

Furthermore, the multi-target nature of natural products makes them ideal candidates for combination therapies [85]. Investigating their synergistic effects with conventional drugs could lead to more effective treatments with reduced side effects.

Ultimately, bridging the gap between promising lab results and patient care requires a commitment to rigorous preclinical and clinical validation. More sophisticated, well-designed, and standardized studies are essential to provide the robust evidence required for regulatory approval and widespread therapeutic use. By embracing these future directions, the scientific community can harness the vast potential of natural products to create a new generation of drugs for ER stress-related diseases. This concerted effort is the crucial next step in translating a historical wealth of knowledge into modern, effective therapies.

**Table 3 ijms-26-08814-t003:** This table provides a concise summary of the key challenges and future directions discussed. It outlines the main hurdles in the research and therapeutic application of natural compounds as ER stress modulators and highlights the innovative solutions and strategies being pursued to overcome them.

Aspect	Challenges	Future Solutions
Natural Compounds	Low bioavailability, variability, complex mechanisms of action.	Semi-synthetic derivatives, advanced delivery systems (nanotechnology).
Research	Poor reproducibility, difficulty isolating specific mechanisms.	High-throughput screening, SAR (structure–activity relationship) studies.
Therapeutic Development	Lack of robust clinical trials and clear regulatory pathways.	Combination therapies, rigorous and standardized clinical validation.

## 6. Conclusions

The evidence compiled in this review highlights the critical role of ER stress in a wide array of human diseases. The dysregulation of the UPR, a central pathological mechanism with a dual capacity for survival and apoptosis, is a key determinant of cell fate, making it a compelling target for therapeutic intervention across conditions ranging from neurodegeneration to cancer and metabolic disorders.

The therapeutic potential of natural products as ER stress modulators is significant. They represent more than just traditional remedies; they are a rich and diverse source of chemical scaffolds for novel drug discovery. While the multi-target nature of these compounds presents a challenge for scientific elucidation, it also provides a unique advantage in clinical practice by offering a holistic approach to complex pathologies. Compounds like curcumin, resveratrol, and sulforaphane exemplify this potential, demonstrating diverse mechanisms for rebalancing the UPR and protecting cells from chronic stress.

In conclusion, the path forward necessitates a renewed commitment to rigorous research. The existing challenges of poor bioavailability, lack of standardization, and a scarcity of clinical validation are substantial but not insurmountable. By leveraging modern scientific tools, including advanced screening technologies, nanotechnology-based delivery systems, and precision medicine, the full therapeutic potential of these compounds can be unlocked. The continued exploration of natural products as ER stress-targeting agents holds the key to a new generation of effective, safe, and potentially more personalized therapies for some of humanity’s most debilitating diseases.

## Figures and Tables

**Figure 1 ijms-26-08814-f001:**
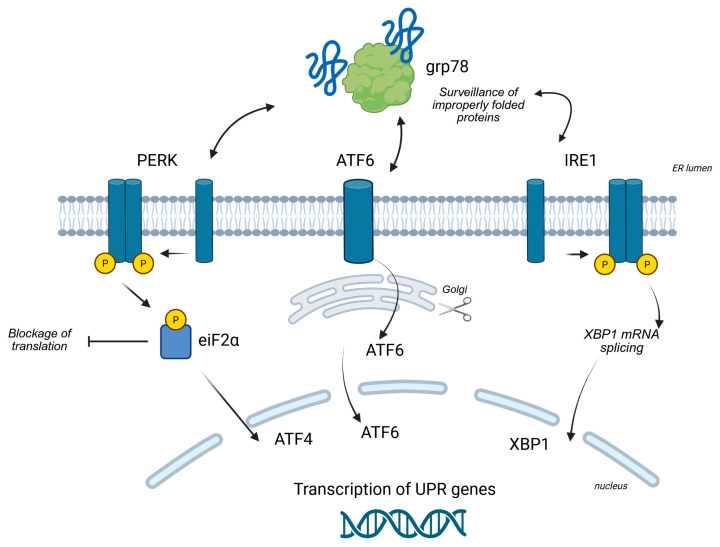
Overview of the UPR signaling branches (see the text for other information). Created in BioRender (https://BioRender.com/9dpx2oz, accessed on 5 September 2025).

## Data Availability

Not applicable.

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
