# Peer review of "Targeting the Unfolded Protein Response with Natural Products: Therapeutic Potential in ER Stress-Related Diseases"

_ijms, 2025, doi:10.3390/ijms26188814_

Round 1

Reviewer 1 Report

Comments and Suggestions for Authors The quoted manuscript, IJMS-3860953 by Elia and Simona, entitled "Targeting the Unfolded Protein Response with Natural Products: Therapeutic Potential in ER Stress-Related Diseases," is a well-written and organised review, providing relevant information about the role of Natural Products against the UPR. The manuscript deserves immediate publication but some points must be modified: please include attractive figures in order to see the flux of info that the authors described. please include the chemical structures of NP cited.

Author Response

The quoted manuscript, IJMS-3860953 by Elia and Simona, entitled "Targeting the Unfolded Protein Response with Natural Products: Therapeutic Potential in ER Stress-Related Diseases," is a well-written and organised review, providing relevant information about the role of Natural Products against the UPR. The manuscript deserves immediate publication but some points must be modified: please include attractive figures in order to see the flux of info that the authors described. please include the chemical structures of NP cited.

We thank the reviewer for positive feedback on our manuscript. We appreciate valuable comments and are delighted that she/he found our work to be an important contribution.

In response to suggestions, we've updated the manuscript with a new figure (Figure 1) and the chemical structures of the molecules discussed. We believe these additions enhance the clarity and overall quality of the paper.

Reviewer 2 Report

Comments and Suggestions for Authors

Reviewer comments and suggestions

This review explores the link between ER stress, the unfolded protein response (UPR), and human disease, emphasizing the therapeutic role of natural products. It categorizes compounds as UPR inhibitors, activators, or indirect modulators that reduce oxidative stress or enhance protein folding. The paper synthesizes current findings to guide research and clinical application. It highlights promising natural products with potential as novel therapeutics for ER stress–related diseases.

Decision: Major revision is needed.
The paper has compiled and presented the therapeutic aspects of natural compounds targeting the unfolded protein response in a commendable manner. However, I recommend incorporating the following points in the revised version of the manuscript:

  1. The abstract should be restructured to follow a more formal format (objective, materials and methods, results, and conclusion). Even though it is a review, the authors can indicate the scope of the literature reviewed and the number of references cited.
  2. No 2025 references were cited. Please include the most recent and relevant research articles as well as review papers.
  3. The manuscript should clearly highlight its novelty compared to existing literature, as there are already numerous articles addressing this theme.
  4. More natural compounds should be included and summarized in a comprehensive table, incorporating details such as animal or human studies, treatment duration, and outcomes.
  5. At least two figures should be added to enhance clarity and meet the standard requirements of the journal.
  6. Sections 3.3 and 3.4 are poorly explained; additional points and elaboration are needed to strengthen these sections.

Author Response

This review explores the link between ER stress, the unfolded protein response (UPR), and human disease, emphasizing the therapeutic role of natural products. It categorizes compounds as UPR inhibitors, activators, or indirect modulators that reduce oxidative stress or enhance protein folding. The paper synthesizes current findings to guide research and clinical application. It highlights promising natural products with potential as novel therapeutics for ER stress–related diseases.

We thank the reviewer for positive feedback on our manuscript. We appreciate valuable comments and are delighted that she/he found our work to be an important contribution.

Decision: Major revision is needed.

The paper has compiled and presented the therapeutic aspects of natural compounds targeting the unfolded protein response in a commendable manner. However, I recommend incorporating the following points in the revised version of the manuscript:

The abstract should be restructured to follow a more formal format (objective, materials and methods, results, and conclusion). Even though it is a review, the authors can indicate the scope of the literature reviewed and the number of references cited.

Thank you for your valuable feedback. We appreciate your suggestion to restructure the abstract to follow a more formal format (objective, materials and methods, results, and conclusion) and to include the scope of literature reviewed and the number of references cited.

However, as per the guidelines for review articles published in IJMS, a formal, structured abstract is not required for this type of manuscript. The standard practice for reviews in this journal is to provide a concise, narrative summary of the topic.

Furthermore, review articles in IJMS are generally not structured into distinct "materials and methods" or "results" sections, as they synthesize existing literature rather than presenting new data. Consequently, a structured abstract would not accurately represent the content or nature of our manuscript.

We believe the current abstract effectively summarizes the scope and key findings of our review, consistent with the journal's format for this type of submission.

 No 2025 references were cited. Please include the most recent and relevant research articles as well as review papers.

We have addressed your suggestion by conducting a comprehensive search for the most recent and relevant literature. We have now updated the manuscript with several new references, including articles published in 2025, to ensure the review is current and reflects the latest advancements in the field. We believe these additions significantly enhance the quality and relevance of our work.

The manuscript should clearly highlight its novelty compared to existing literature, as there are already numerous articles addressing this theme.

Thank you for the constructive feedback. We have revised the introduction to more explicitly highlight the novelty of our work compared to the existing literature.

More natural compounds should be included and summarized in a comprehensive table, incorporating details such as animal or human studies, treatment duration, and outcomes.

In the revised manuscript, we have addressed the reviewers' concerns regarding the inclusion of additional natural compounds. We have expanded our coverage to include several other compounds, each with a dedicated section detailing their specific interactions with the UPR and ER stress. To enhance clarity and scientific rigor, we have also incorporated the chemical structures of these compounds within the manuscript.

We have opted to present a concise summary of the effects of these compounds in a table, as requested. However, instead of including extensive details on animal or human studies and treatment duration within the table, we have decided to present the salient information: the compound's source, and the primary mechanism of action. This approach ensures that the table remains a clear and easy-to-reference summary

To avoid redundancy and maintain the flow of the main text, we have chosen to reserve the more detailed information on in vitro and in vivo studies, including specifics on experimental models, dosages, and treatment durations, for the updated full-length text.

At least two figures should be added to enhance clarity and meet the standard requirements of the journal.

Thank you for your constructive feedback. In response to your suggestion, we have added a new figure to the manuscript to enhance the clarity of the signaling pathways discussed.

We would also like to point out that the manuscript already includes two tables that summarize the key data from our study, which we believe further contributes to a clear and organized presentation that meets the journal's standards.

Sections 3.3 and 3.4 are poorly explained; additional points and elaboration are needed to strengthen these sections.

We agree with the reviewer's feedback that sections 3.3 and 3.4 required more detailed elaboration. To address this, we have significantly expanded both sections to provide a more comprehensive explanation of the roles of ER stress and the UPR in metabolic, inflammatory, and autoimmune diseases. We believe these additions provide the necessary depth and a stronger scientific foundation for these sections, significantly improving the overall quality of the ms.

Reviewer 3 Report

Comments and Suggestions for Authors

The manuscript by Elia Ranzato, entitled “Targeting the Unfolded Protein Response with Natural Products: Therapeutic Potential in ER Stress-Related Diseases,” provides a comprehensive review of UPR pathways and their connection to various diseases. The authors also describe the role of natural products in treating these conditions. The manuscript is well-written, and incorporating the minor suggestions listed below will further enhance its clarity and overall quality.

  1. Throughout the manuscript, there are several very short paragraphs (only two to three lines long). I suggest revising them into medium-sized paragraphs to improve readability and flow.
  2. I recommend including a representative figure illustrating the UPR pathway.
  3. I recommend including the chemical structures of the natural products mentioned in the manuscript to enhance clarity.
  4. I recommend dividing the natural products discussed in Section 4.2 (key examples of natural products and their effects) into subsections to improve organization and enhance readability.
  5. I recommend including a section on ER stress in respiratory diseases to provide a more comprehensive overview.

Author Response

The manuscript by Elia Ranzato, entitled “Targeting the Unfolded Protein Response with Natural Products: Therapeutic Potential in ER Stress-Related Diseases,” provides a comprehensive review of UPR pathways and their connection to various diseases. The authors also describe the role of natural products in treating these conditions. The manuscript is well-written, and incorporating the minor suggestions listed below will further enhance its clarity and overall quality.

We thank you the reviewer for positive feedback on our manuscript. We appreciate the valuable comments and are delighted that she/he found our work to be an important contribution.

In response to suggestions, we've updated the manuscript. We believe these additions enhance the clarity and overall quality of the paper.

1. Throughout the manuscript, there are several very short paragraphs (only two to three lines long). I suggest revising them into medium-sized paragraphs to improve readability and flow.

We thank the reviewer for the excellent feedback regarding paragraph length and readability. We agree that a more consistent paragraph structure will improve the manuscript's flow and overall quality.

We believe the revisions have significantly enhanced the readability of the text, making it easier for the reader to follow the scientific arguments and the flow of the manuscript.

We are confident that this change addresses the reviewer's concern and strengthens the quality of the paper

2. I recommend including a representative figure illustrating the UPR pathway.

In response to your recommendation, we have inserted a representative figure illustrating the UPR pathway into the manuscript.

3. I recommend including the chemical structures of the natural products mentioned in the manuscript to enhance clarity.

We have inserted chemical structures

4. I recommend dividing the natural products discussed in Section 4.2 (key examples of natural products and their effects) into subsections to improve organization and enhance readability.

We appreciate the reviewer's feedback regarding the organization of Section 4.2. We agree that dividing the natural products into subsections will significantly improve the chapter's structure and readability.

To address this, we have revised Section 4.2 and have categorized the compounds based on their primary chemical class and established therapeutic focus. The new subheadings are:

Flavonoids and Polyphenols: This subsection will group compounds such as Curcumin, Resveratrol, EGCG, Quercetin and Oleuropein which are all polyphenolic in nature and known for their antioxidant and anti-inflammatory properties.

Alkaloids and Saponins: This category will include Berberine, Piperine, Ginsenosides, and Astragaloside IV, highlighting their diverse pharmacological actions.

Terpenoids and Other Compounds: This final subsection will cover Camphene, Honokiol, Ursolic Acid and Sulphorane which belong to the terpenoid family or are structurally distinct.

5. I recommend including a section on ER stress in respiratory diseases to provide a more comprehensive overview.

We agree with the reviewer's suggestion that a dedicated section on ER stress in respiratory diseases would enhance the comprehensiveness of our review. To address this, we have added a new section, Section 3.6: Respiratory Diseases.

Round 2

Reviewer 2 Report

Comments and Suggestions for Authors

All comments have been addressed.